# From Interaction Trajectories to Prompt Rules: Credit Assignment for Multi-Agent Prompt Optimization

**Bin Wu** [1]  **Haoran Xu** [2]  **Xiang Zhuang** [3]  **Zonghao Chen** [1]  **Zhu Li** [4]
**Emine Yilmaz** [1]  **Qiang Zhang** [2]

## Abstract

Large language model (LLM)-based multi-agent systems commonly rely on natural-language prompts to specify agent behavior, yet optimizing these prompts remains challenging when agent roles and interaction structures are fixed by design. In such systems, **behaviors emerge over long, noisy interaction trajectories, making it difficult to determine which prompt components are responsible for success or failure**. As a result, outcome-level feedback alone is insufficient, while existing prompt optimization methods typically rely on final task scores or global prompt rewrites, limiting their ability to exploit trajectory evidence or support the localized updates. We propose Trajectory-based Rule Credit Estimation (TRUCE), a framework for prompt optimization in multi-agent systems that explicitly addresses this credit assignment challenge. TRUCE performs trajectory-aware attribution by linking outcome feedback to informative sub-trajectories and translating the resulting credit signals into unit-level edits over prompt-defined behavioral rules. By preserving agent roles and interaction structures, TRUCE enables prompt refinement through localized updates aggregated across tasks. Experiments on multiple benchmarks demonstrate that TRUCE consistently improves task performance and efficiency over competitive baselines. Code is available at `https://github.com/bingo-w/TRUCE`.

## 1. Introduction

The rapid progress of large language models (LLMs) (Brown et al., 2020) has shifted AI from isolated, single-model systems toward ecosystems of interacting autonomous agents. In LLM-based multi-agent systems (MAS) (Guo et al., 2024), agents perceive complex environments (Li et al., 2024b), communicate via natural language, and coordinate over extended horizons to solve tasks that exceed the capability of any individual agent (Zhu et al., 2025). Such systems have shown promise across diverse domains, including collaborative software engineering (Qian et al., 2023) and clinical decision-making (Li et al., 2024a). In most existing MAS deployments, however, agent roles and interaction protocols are fixed at design time, unlike recent structure-optimization frameworks that alter agent connectivity (Zhuge et al., 2024). Natural-language prompts therefore become the primary and often the only mechanism for shaping agent behavior. In effect, prompts act as *behavioral policies*: small ambiguities or mis-specifications can propagate through long-horizon interactions, leading to inefficiency or failure, whereas carefully designed prompts can induce coherent and effective collective behavior.

In practice, improving multi-agent performance through prompts remains a highly manual and expert-driven process (see Figure 1). When a system underperforms, human experts inspect *interaction trajectories*, including reasoning traces, message exchanges, and intermediate actions, to diagnose behavioral failures (Zhuge et al., 2025). Rather than rewriting prompts wholesale, experts typically apply *minimal, localized, unit-level refinements*: clarifying a responsibility, adding a missing constraint, or relaxing an overly restrictive instruction. This trajectory-informed and minimal-edit refinement strategy is often effective, but it relies heavily on human expertise and does not scale with growing agent populations or interaction complexity.

To reduce manual effort, recent work has explored automated prompt optimization. A prominent line of research uses a black-box process, updating prompts based on input–output pairs or final scores (Khattab et al., 2024; Zhuge et al., 2024). In multi-agent settings, however, outcome-level feedback is fundamentally insufficient: final results

[1]Centre for Artificial Intelligence, University College London, London, United Kingdom [2]Zhejiang University, Hangzhou, China [3]Shanghai AI Laboratory, Shanghai, China [4]Mingdu Tech, Hangzhou, China. Correspondence to: Qiang Zhang <qiang.zhang.cs@zju.edu.cn>.

*Proceedings of the 43rd International Conference on Machine Learning*, Seoul, South Korea. PMLR 306, 2026. Copyright 2026 by the author(s).

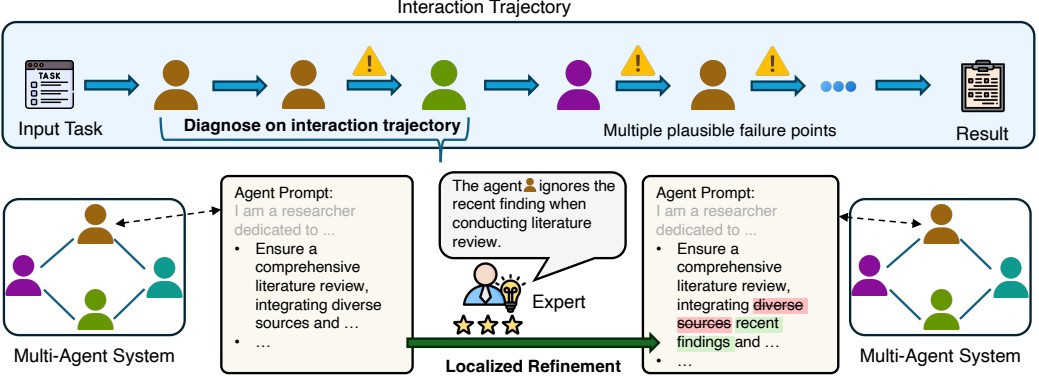

*Figure 1.* Expert-driven prompt refinement in multi-agent systems. Experts analyze interaction trajectories and apply localized updates to agent prompt rules. The key challenge is credit assignment from trajectory evidence to specific prompt components.

conflate the effects of many interdependent interaction steps. A successful outcome does not reveal whether agents coordinated efficiently or redundantly (Wu et al., 2025), while a failure provides little guidance about which agent, instruction, or interaction stage was responsible (Zhang et al., 2025c). Consequently, such methods often resort to coarse or generic prompt updates (Cemri et al., 2025).

More recent reflection-based approaches leverage trajectory-level information through end-to-end critiques or textual gradients (Yuksekgonul et al., 2025; Cheng et al., 2024). While these methods incorporate richer execution evidence, they typically operate on prompts as *monolithic text*. This global refinement paradigm obscures the relationship between specific prompt components and observed behaviors, and it prevents minimal, targeted updates, an essential property in multi-agent systems where agent behaviors are tightly coupled and interact over time (Yin & Wang, 2025). As a result, existing approaches fail to jointly capture two defining characteristics of expert prompt refinement: (i) using interaction trajectories as diagnostic evidence, and (ii) translating evidence into minimal, localized prompt updates.

These limitations point to a central challenge in automated prompt optimization for multi-agent systems: *credit assignment*. Prompts influence behavior indirectly through long, evolving interaction trajectories, rather than through explicit action selection. The effect of any individual prompt instruction is mediated by rich and redundant interaction context, making it difficult to determine which parts of a trajectory reflect the influence of which prompt components. This fundamental challenge is further amplified by inter-agent dependencies and delayed effects across multiple interaction rounds. Without an explicit mechanism for assigning trajectory-level evidence to specific prompt units, scalable and reliable prompt optimization remains elusive.

To address this challenge, we introduce **Trajectory-based Rule Credit Estimation (TRUCE)**, a framework for

prompt optimization in LLM-based multi-agent systems. TRUCE is grounded in a structural analogy to reinforcement learning (RL), instantiated entirely in natural language space. In this formulation, prompts correspond to policies, interaction trajectories to execution rollouts, and task evaluations to outcome feedback. In this analogy, interpretable prompt rules correspond to policy units that can be refined independently. Unlike traditional RL, where numeric rewards are propagated through differentiable parameters, TRUCE performs *verbalized credit assignment*: it attributes outcome feedback to informative sub-trajectories and produces localized, interpretable credit signals for individual prompt rules. These signals are then translated into unit-level policy-editing suggestions, enabling minimal and targeted prompt refinement. By aggregating updates across tasks, TRUCE stabilizes optimization and reduces sensitivity to individual trajectories. This allows TRUCE to inherit the conceptual structure of RL, including policy, rollout, credit assignment, and policy update, while adapting it to the constraints of prompt-based multi-agent systems. Table 1 summarizes the correspondence between traditional RL components and their counterparts in TRUCE.

Our contributions are fourfold: (1) We formulate prompt optimization in multi-agent systems as a trajectory-based, unit-based problem, providing a structured alternative to monolithic prompt refinement. (2) We propose a trajectory-based credit assignment mechanism that extracts informative signals from long and noisy multi-agent interaction traces. (3) We introduce a unit-level prompt representation and structured policy-editing procedure that enables minimal, localized prompt updates. (4) We empirically validate TRUCE on collaborative and competitive multi-agent benchmarks, demonstrating consistent improvements in task performance and coordination efficiency.

| Aspect | Traditional RL | TRUCE | Example - TRUCE |
|---|---|---|---|
| **Policy** | Numeric parameters (e.g., neural network weights) optimized via gradient-based methods. | Natural language prompts (e.g., behavior rules, or task guideline) optimized in verbalized (non-gradient) way | *Encourage comprehensive literature reviews and integration of recent findings into research proposal* |
| **Credit Assignment** | Numeric proxy, like value estimation and advantage functions. | Verbalized credit assignment derived from outcome feedback based on sub-trajectory. | *The current iteration makes significant progress towards task completion by defining the research question and developing a detailed methodology...* |
| **Update Mechanism** | Policy gradient derived from numeric credit. | Policy-editing suggestion derived from verbalized outcome and process feedback. | *Modify literature review prompts to ensure comprehensive coverage and integration of recent advancements* |

*Table 1.* Comparison between traditional reinforcement learning (RL) and our proposed TRUCE.

## 2. Related Works

**LLM-Based Multi-Agent Systems: Structure vs. Prompted Behavior.** LLM-based multi-agent systems (MAS) have emerged as a powerful paradigm for complex tasks requiring coordination and iterative reasoning, with demonstrated success in software development (Qian et al., 2023) and medical diagnosis (Li et al., 2024a). In these systems, multiple LLM agents interact via natural language and assume distinct roles. Prior work emphasizes either structured role-based workflows or dynamic communication strategies, including debate (Khan et al., 2024) and iterative self-reflection (Madaan et al., 2023). Broadly, research on improving LLM-based MAS follows two directions: *system-level structure optimization*, such as automated role assignment, communication protocols, and interaction workflows (Hu et al., 2024; Zhang et al., 2024; Zhuge et al., 2024), and *behavioral specification*, where agent behaviors and coordination strategies are encoded directly in natural-language prompts (Zhang et al., 2025b;a). In many practical deployments, particularly those built on frozen or proprietary LLMs, system structure and agent roles are fixed or costly to redesign. In such settings, prompt refinement becomes the primary mechanism for improving system behavior, with prompts functioning as implicit behavioral policies that govern long-horizon interactions. This work focuses on this latter setting.

**Prompt Optimization beyond Black-Box and Global Rewrites.** Prompt optimization has progressed from early gradient-based tuning methods (Lester et al., 2021) to approaches that leverage LLMs as optimizers. A prominent line of work relies on *numeric or scalar feedback*, framing prompt optimization as program synthesis or search guided by evaluation metrics (Yang et al., 2024; Fernando et al., 2024; Khattab et al., 2024). These methods treat prompts as black boxes and optimize them based on input–output performance; while effective in single-step or single-agent settings, such outcome-driven optimization provides limited diagnostic insight in multi-agent systems, where final results conflate many interdependent interaction steps. Another line of research adopts reflection-based optimization, incorpo-

rating *richer verbal feedback* such as natural-language critiques or textual gradients (Shinn et al., 2023; Pryzant et al., 2023; Yuksekgonul et al., 2025). Although more expressive, these approaches usually operate end-to-end over long contexts, treating prompts as monolithic text and producing global, hard-to-localize revisions that limit effectiveness in tightly coupled multi-agent settings. Recent efforts distill prior experience into reusable natural-language policies, including ExpeL (Zhao et al., 2024), AutoGuide (Fu et al., 2024), Mobile-AgentE (Wang et al., 2025), and GiGPO (Feng et al., 2025). While these approaches highlight the value of experience-based and policy-centric representations, they are primarily designed for single-agent scenarios or inference-time guidance, and do not address iterative, trajectory-driven prompt refinement in multi-agent systems.

**Trajectory-Level Credit Assignment and Unit-Based Policy Revision.** Understanding how intermediate decisions contribute to final outcomes is a longstanding challenge in sequential and agentic systems. LLM-based agents typically solve tasks through multi-step interactions (Yao et al., 2023), yet many benchmarks emphasize final success rates, which are often an unreliable proxy for capability in multi-agent settings, overlooking robustness, efficiency, and coordination quality (Zhuge et al., 2025). Recent work has therefore focused on trajectory-level analysis, where interaction histories reveal reasoning chains, coordination dynamics, and failure modes (Lù et al., 2025), supporting error attribution (Zhang et al., 2025c) and root-cause diagnosis (Cemri et al., 2025). In parallel, reinforcement learning formalizes this challenge as *credit assignment*, attributing delayed rewards to earlier actions via mechanisms such as value estimation and advantage functions (Jia & Zhou, 2022; Schulman et al., 2015). Separately, prior work has explored policy-centric or unit-based representations that distill experience into reusable natural-language rules or guidelines (Zhao et al., 2024; Fu et al., 2024; Wang et al., 2025). While these efforts highlight the importance of both trajectory-based credit signals and interpretable policy units, they are typically studied in isolation. **How to systematically integrate trajectory-based credit assignment with unit-level policy revision for prompt optimization in multi-agent systems remains**

**an open problem, which this work aims to address**.

## 3. Preliminary

### 3.1. Prompt Optimization in LLM-based MAS

Let an LLM-based multi-agent system (MAS) consist of a set of agents $\mathcal{A} = \{a_i\}_{i=1}^N$, where each agent $a_i$ is powered by a fixed LLM and guided by a natural-language prompt $P_i$. The prompt specifies the agent's role and provides natural-language instructions that shape how it reasons, communicates, and responds during interaction. Given a task $\mathcal{T}$, agents interact over multiple steps through natural language, exchanging messages and producing intermediate reasoning and actions to generate a final response $r$. The execution induces an interaction trajectory $\mathcal{H}_T = (h_1, \ldots, h_T)$, where each step captures the relevant interaction context.

We focus on settings where the underlying LLMs, agent roles, and interaction structure are fixed, as is common in practical deployments with frozen or proprietary models. Under these constraints, prompt refinement is the primary mechanism for improving system behavior: prompts function as behavioral policies that condition agent actions throughout the interaction. Given a task-specific evaluator $R(\mathcal{T}, r)$, our objective is to improve expected performance over a task distribution $\mathcal{D}$ by refining the prompts:

$$\max_{\{P_i\}} \mathbb{E}_{\mathcal{T} \sim \mathcal{D}}[R(\mathcal{T}, r)], \quad \text{s.t. } r = \text{MAS}(\mathcal{T}; \{P_i\}). \quad (1)$$

### 3.2. Credit Assignment as a Core Challenge

A central challenge in prompt optimization for LLM-based multi-agent systems is *credit assignment*: determining how responsibility for a final outcome should be attributed to intermediate interaction segments and, ultimately, to the prompt components that governed agent behavior, given an interaction trajectory $\mathcal{H}_T$ and an outcome evaluation $R(\mathcal{T}, r)$. This challenge stems from fundamental structural properties of multi-agent execution. Interaction trajectories are *long, noisy, and highly coupled across agents and time*, with behaviors influencing downstream interactions in non-local ways. As a result, trajectories contain rich but redundant process-level evidence, making it difficult to isolate which steps or instructions caused success or failure. At the same time, prompts are high-level and semantically entangled, and LLM behavior is sensitive to wording, where small local changes can induce large, non-local effects.

Consequently, outcome-level feedback alone is insufficient for guiding prompt refinement in long-horizon, multi-agent settings. Prompts influence behavior indirectly through evolving interaction context and interdependent agent actions, fundamentally obscuring the link between observed outcomes and the prompt instructions that generated them. Systematically connecting trajectory-based evidence to lo-calized prompt updates is therefore essential for stable and scalable prompt optimization.

## 4. Methodology

### 4.1. Overview and Design Principles

To address the credit assignment challenge in prompt optimization for LLM-based multi-agent systems, we propose **Trajectory-based Rule Credit Estimation (TRUCE)**. TRUCE leverages interaction trajectories to attribute outcome feedback to specific behavioral components within agent prompts and uses this attribution to guide stable and interpretable prompt refinement. Unlike black-box prompt search, TRUCE explicitly reasons about how agent behaviors unfold during execution, enabling targeted updates grounded in trajectory-level evidence. In our implementation, all components of TRUCE, including trajectory analysis, credit attribution, policy-editing suggestion generation, and aggregation, are implemented using LLMs guided by structured natural-language prompts.

TRUCE operates in an iterative optimization loop: the system is executed to produce an interaction trajectory and outcome evaluation, which are analyzed to generate structured, natural-language policy-editing suggestions. These suggestions are aggregated across trajectories and tasks to produce minimal, coherent prompt updates, progressively improving system behavior while avoiding destabilizing global rewrites. TRUCE follows three core principles: **trajectory-based credit assignment**, which leverages execution traces beyond outcome-level feedback; **unit-based refinement**, which treats prompts as collections of interpretable behavioral rules; and **minimal, interpretable updates**, which express refinements as localized natural-language edits that mirror expert practice. These principles operationalize the credit assignment formulation in Section 3.

### 4.2. Trajectory-Based Credit Assignment

The first step is to attribute outcome-level feedback to the interaction process. Given a task $\mathcal{T}$, a set of agent prompts $\{P_i\}$, and an execution trajectory $\mathcal{H}_T = (h_1, \ldots, h_T)$ with outcome evaluation $R(\mathcal{T}, r)$, the goal is to identify which parts of the trajectory contributed most significantly to the final outcome. Rather than treating the trajectory as an undifferentiated context, TRUCE seeks localized attribution that preserves temporal and agent-level structure.

Multi-agent trajectories are long, noisy, and highly coupled across agents and time, making direct reasoning over the full trajectory impractical. To address this, TRUCE partitions the trajectory into a set of coherent sub-trajectories $\{h^{(k)}\}$, where each sub-trajectory corresponds to a protocol-level interaction unit that captures a semantically coherent stage of multi-agent execution. In practice, these units are de-

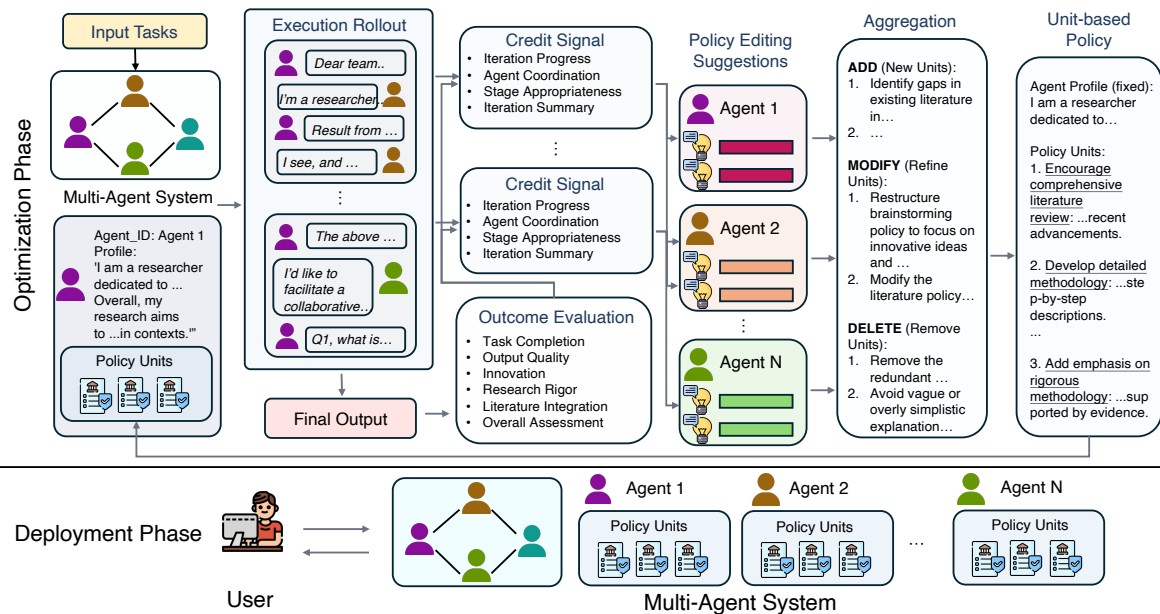

*Figure 2.* **Overview of TRUCE**. Outcome feedback is attributed to sub-trajectories to generate and aggregate unit-level policy edits, enabling iterative prompt refinement and deployment to unseen tasks.

fined using the native execution structure of the underlying system, such as communication rounds in iterative interaction protocols (Zhu et al., 2025), or task-stage blocks in structured workflows (Qian et al., 2023). This segmentation follows fixed execution boundaries rather than post-hoc heuristics, ensuring reproducible localized attribution.

For each sub-trajectory $h^{(k)}$, TRUCE estimates its contribution to the final outcome by jointly considering the interaction segment and the outcome evaluation. Formally, we associate each sub-trajectory $h^{(k)}$ with a *verbalized (natural-language) credit signal*:

$$c^{(k)} = \Phi\big(h^{(k)}, R(\mathcal{T}, r)\big), \qquad (2)$$

where $\Phi(\cdot)$ is a trajectory-aware attribution function. In practice, $\Phi(\cdot)$ is implemented as an LLM that analyzes sub-trajectories and outcome feedback via structured prompts to produce natural-language credit signals. The resulting credit signal $c^{(k)}$ qualitatively characterizes whether the behaviors in $h^{(k)}$ supported or hindered task completion. These signals transform a single outcome-level evaluation into localized, interpretable attribution over the interaction trajectory and provide the foundation for unit-level prompt refinement in subsequent stages.

### 4.3. Unit-Based Prompt Representation

To translate trajectory-level credit into actionable prompt updates, TRUCE adopts a unit-based representation of prompts. In our setting, agent roles and interaction structure are fixed and define the system topology; optimization

therefore operates only on the behavioral rules that guide how each agent fulfills its role during execution.

Concretely, TRUCE maintains a collection of *policy units* $P_i = \{u_{i,1}, \dots, u_{i,M_i}\}$ for each agent, where each unit corresponds to a semantically coherent behavioral rule expressed in natural language. These units serve as modular optimization abstractions for localized refinement rather than requiring global prompt rewrites. The original prompt context (e.g., role definitions and base instructions) is preserved, while policy units are dynamically added, revised, or removed to refine agent behavior over optimization.

Importantly, policy units are not assumed to have an explicit or predefined alignment with interaction segments. Instead, TRUCE begins by identifying sub-trajectories that contribute positively or negatively to the final outcome and then analyzes the agent behaviors exhibited in those segments to infer which policy units, or which parts of policy units, are implicated. This attribution enables localized and interpretable prompt refinement even when the relationship between prompts and behavior is indirect or entangled.

### 4.4. Policy-Editing Suggestion Generation

Given trajectory-based credit assignment and the current agent prompts, TRUCE translates attribution into actionable prompt updates by generating a set of *policy-editing suggestions*. Each individual suggestion $\delta$ is represented as a tuple $(\tau, s)$, where $\tau \in \{\text{ADD}, \text{MODIFY}, \text{DELETE}\}$ denotes the edit type and $s$ is a natural-language instruction describing the proposed change. Suggestions may target an existing

policy unit, a subset of a policy unit, or introduce a new behavioral rule when effective behaviors are observed but not explicitly specified. Formally, TRUCE generates a set of policy-editing suggestions

$$\Delta = \{\delta\} = \Psi\big(\{c^{(k)}\}_k, \{P_i\}\big), \qquad (3)$$

where $\Psi(\cdot)$ maps trajectory-level credit signals and the current prompts to a collection of policy-editing proposals. The function $\Psi(\cdot)$ is realized by an LLM that conditions on trajectory-level credit signals and the current prompts to generate structured policy-editing suggestions. Each suggestion implicitly operates on the unit-based prompt representation introduced in Section 4.3, by refining an existing policy unit or by introducing a new policy unit when necessary.

Modify suggestions refine the scope or emphasis of existing behavioral rules whose partial application contributed to suboptimal behavior; delete suggestions remove rules or rule components that repeatedly lead to undesirable outcomes; and add suggestions introduce missing behavioral rules revealed by successful trajectory segments. Importantly, all suggestions operate strictly on behavioral rules and do not alter agent roles or interaction structure, ensuring alignment with the problem setting defined in Section 3.

### 4.5. Aggregated Prompt Refinement Across Tasks

Policy-editing suggestions generated from individual trajectories are inherently local and may reflect task-specific artifacts. To ensure robustness and generalization, TRUCE aggregates suggestions across multiple trajectories and tasks before applying prompt updates. Given suggestion sets $\{\Delta^{(\mathcal{T})}\}$ generated from tasks sampled from $\mathcal{D}$, TRUCE first measures the support of each suggestion $\delta$ by how frequently it appears across tasks,

$$\text{supp}(\delta) = \mathbb{E}_{\mathcal{T}\sim\mathcal{D}}\left[\mathbf{1}\{\delta \in \Delta^{(\mathcal{T})}\}\right], \qquad (4)$$

and retains the top-$k$ suggestions with the highest support,

$$\Delta_{\text{final}} = \text{TopK}_\delta \ \text{supp}(\delta). \qquad (5)$$

Suggestion aggregation and the subsequent application of retained edits are also performed by LLMs, which reconcile semantically similar suggestions and apply localized refinements to the prompts.

Only retained suggestions are applied as minimal edits to the unit-based prompt representation, refining existing policy units or adding new ones as needed. This aggregation step suppresses spurious updates arising from individual trajectories, stabilizes optimization under prompt sensitivity, and yields behavioral improvements that generalize across tasks, completing the trajectory-based prompt refinement loop. The retained suggestions $\Delta_{\text{final}}$ are applied as localized edits to the agent prompts, yielding an updated prompt set for the next optimization iteration.

### 4.6. Theoretical Intuition and Analysis

**Intuition.** TRUCE can be viewed as a trajectory-grounded local optimization procedure over prompt-defined behavioral rules. By representing prompts as collections of interpretable policy units, TRUCE restricts optimization to localized edits that modify only a small part of an agent's behavior. Trajectory-based credit assignment further constrains this local search space by identifying unit-level edits repeatedly implicated by execution evidence, while aggregation across tasks suppresses spurious updates. These mechanisms guide optimization toward small, evidence-supported changes, reducing the risk of destabilizing global prompt rewrites and enabling stable improvement in practice.

**Formal perspective.** This intuition can be formalized by viewing TRUCE as optimizing the expected task objective $J(P) = \mathbb{E}_{\mathcal{T}\sim\mathcal{D}}[R(\mathcal{T}, P)]$, over a discrete space of unit-level prompt edits. Trajectory-based credit assignment induces an evidence-guided neighborhood around the current prompt, and aggregation acts as an acceptance filter that favors edits with consistent support across tasks. A formal analysis with convergence guarantees is in Appendix C.

## 5. Experimental Setup

**Benchmark and Evaluation.** We evaluate on two benchmarks: *MultiAgentBench* (Zhu et al., 2025) and the *Programming* benchmark (Islam et al., 2024; 2025). For *MultiAgentBench*, we consider three collaborative domains (*Research*, *Coding*, *Database*) and one competitive domain (*Bargaining*), and report **Task Score** and **Coordination Score** following the original evaluation protocol. For the *Programming* benchmark, we evaluate on four datasets—*HumanEval*, *HumanEval-ET*, *MBPP*, and *MBPP-ET* using the state-of-the-art MAS framework CODESIM (Islam et al., 2025), and report Pass@1. All datasets are split into disjoint train/validation/test sets. Additional details are in Appendix B.

**Baselines.** We compare TRUCE with: (1) Original: the prompts provided by the benchmark. (2) Reflexion (Shinn et al., 2023): which involves the verbalized feedback into the prompt. (3) DsPy (MIPROv2) (Opsahl-Ong et al., 2024): which evolves prompts using numeric feedback. (4) TextGrad (Yuksekgonul et al., 2025): which uses textual gradients as a proxy for optimization. (5) MAPRO (Zhang et al., 2026): which optimizes multi-agent prompts via maximum a posteriori inference. (6) HiveMind (Xia et al., 2026): which optimizes multi-agent prompts through contribution-guided online updates.

**Model and Prompt.** We evaluate two model families: `gpt-4o` and `Qwen3`. Specifically, for `gpt-4o`, we use `gpt-4o-mini-2024-07-18` as the task-executing model and `gpt-4o-2024-08-06` as the optimizer. For `Qwen3`, we use `Qwen/Qwen3-32B` and

`Qwen/Qwen3-235B-A22B-Instruct-2507` for task execution and optimization, respectively. For evaluation on *MultiAgentBench*, we use `gpt-4o-2024-08-06` as the LLM-as-a-judge. For both TRUCE and all baselines, prompts are optimized on the training set, selected on the validation set, and evaluated on the test set, with at most five optimization rounds (see Appendix B.3).

# 6. Result Analysis

We evaluate our method from four perspectives: overall performance on *MultiAgentBench* (Section 6.1), generalization to a state-of-the-art multi-agent programming system (Section 6.2), execution efficiency (Section 6.3), and ablation studies isolating the contributions of trajectory-level credit assignment and unit-based prompt refinement (Section 6.4).

## 6.1. Overall Performance on *MultiAgentBench*

Table 2 reports results on *MultiAgentBench* across three collaborative domains and one competitive domain. TRUCE consistently outperforms strong generic prompt optimization baselines in both task score (TS) and coordination score (CS), demonstrating the effectiveness of trajectory-based rule credit estimation for multi-agent prompt optimization. Compared to DsPy, which relies on scalar outcome feedback, TRUCE achieves higher performance across all domains, highlighting the limitations of outcome-only optimization in long-horizon, interdependent multi-agent settings. TRUCE also consistently outperforms TextGrad, indicating the advantage of unit-level prompt refinement over global prompt rewrites.

Performance gains differ systematically across settings. In collaborative domains (*Research*, *Coding*, and *Database*), TRUCE yields substantial improvements in coordination score, reflecting more effective communication, role adherence, and planning among agents. In the competitive *Bargaining* domain, gains are more modest but consistent, suggesting that trajectory-based refinement remains beneficial even under partially misaligned objectives. Overall, coordination scores exhibit larger relative improvements than task scores, indicating that trajectory-based credit assignment primarily enhances interaction dynamics, which in turn improves task outcomes. These improvements are also stable across multiple random seeds: on *Research*, TRUCE achieves TS of $83.13 \pm 0.50$ and CS of $89.35 \pm 0.46$, while on *Bargaining*, it achieves TS of $82.55 \pm 0.80$ and CS of $83.50 \pm 0.64$ (Appendix D.2).

We additionally compare TRUCE with two recent multi-agent prompt optimization methods, MAPRO and Hive-Mind, on two collaborative domains (*Research* and *Coding*) under the GPT-4o-mini setting. As shown in Table 4, TRUCE consistently outperforms both methods, indicating

that trajectory-aware credit assignment with localized unit-level refinement provides stronger optimization signals than existing multi-agent prompt optimization strategies.

## 6.2. Results on *Programming* Benchmarks

Table 3 reports results on a state-of-the-art multi-agent coding system (Islam et al., 2025). Prompt optimization in this setting is challenging, as the system is carefully engineered and its prompts are manually designed by domain experts. As a result, existing prompt optimization methods show little to no improvement over the original system.

Despite this difficulty, our method consistently achieves performance gains across most datasets. By leveraging trajectory-level evidence and applying localized, unit-based edits, our approach identifies residual optimization opportunities that are inaccessible to outcome-only or global prompt updates. Importantly, even when baseline performance is near saturation, our method does not cause degradation, indicating stable optimization behavior. These results suggest that credit assignment remains valuable even for expert-designed multi-agent systems, enabling systematic refinement without modifying agent roles or interaction structure.

## 6.3. Efficiency Analysis

Beyond final task performance, effective prompt optimization should also improve execution efficiency, as inefficient coordination and redundant interactions are common in long-horizon multi-agent systems. We evaluate efficiency using milestones defined in *MultiAgentBench*, which quantify intermediate task progress.

As shown in Figure 3, our method consistently achieves more milestones per interaction round and per million tokens across all domains, indicating more effective progress under identical execution budgets. Consistent trends are observed when tracking accumulated milestones over interaction rounds or token usage (Figure 4), where our method exhibits steeper slopes than all baselines, reflecting faster task progress throughout execution. Appendix B.4 provides a detailed discussion on offline optimization cost.

## 6.4. Ablation Study

We conduct ablation studies to isolate the contributions of trajectory-based credit assignment and unit-based prompt refinement (Figure 5). Removing credit assignment and relying only on final outcome feedback substantially degrades performance, in some cases below the unoptimized baseline, indicating that outcome-level rewards alone are insufficient for effective unit-level refinement.

When unit-based refinement is removed but trajectory-level credit is preserved, performance improves relative to no

| | collaborative | | | | | | competitive | | | |
| | Research | | Coding | | Database | | Bargaining | | Average | |
| | TS | CS | TS | CS | TS | CS | TS | CS | TS | CS |
|---|---|---|---|---|---|---|---|---|---|---|
| | gpt-4o-mini + gpt-4o | | | | | | | | | |
| Original | 80.00 | 84.69 | 51.00 | 63.26 | 47.33 | 89.33 | 75.09 | 83.82 | 63.36 | 80.28 |
| Reflexion | 79.33 | 86.10 | 49.33 | 63.04 | 44.67 | 91.00 | 80.33 | 83.15 | 63.42 | 80.82 |
| DsPy (MIPROv2) | 82.33 | 82.95 | 50.67 | 64.00 | 54.67 | 93.90 | 55.00 | 78.38 | 60.67 | 79.81 |
| TextGrad | 80.33 | 86.25 | 51.00 | 67.62 | 56.00 | 88.65 | 63.67 | 78.33 | 62.75 | 80.21 |
| TRUCE | **83.00** | **88.50** | **52.00** | **85.54** | **58.67** | **94.00** | **82.33** | **85.03** | **69.00** | **88.27** |
| | qwen-3-32b + qwen-3-235b | | | | | | | | | |
| Original | 83.67 | **88.25** | 53.75 | 77.40 | 66.67 | 91.60 | 83.06 | 81.74 | 71.79 | 84.75 |
| Reflexion | 85.09 | 86.96 | 54.50 | 69.47 | 68.00 | 87.67 | 90.27 | 82.43 | 74.47 | 81.63 |
| DsPy (MIPROv2) | 83.33 | 87.61 | 55.75 | 77.00 | 64.00 | 87.00 | 85.00 | 82.57 | 72.02 | 83.55 |
| TextGrad | 85.26 | 83.33 | 56.25 | 70.60 | 68.00 | 88.00 | 87.78 | 77.25 | 74.32 | 79.80 |
| TRUCE | **86.33** | 86.75 | **57.00** | **79.07** | **70.00** | **94.80** | **93.08** | **83.21** | **76.61** | **85.96** |

*Table 2.* Average Task Score (TS), Coordinate Score (CS) on three collaborative domains: *Research*, *Coding*, and *Database*, and one competitive domain: *Bargaining*.

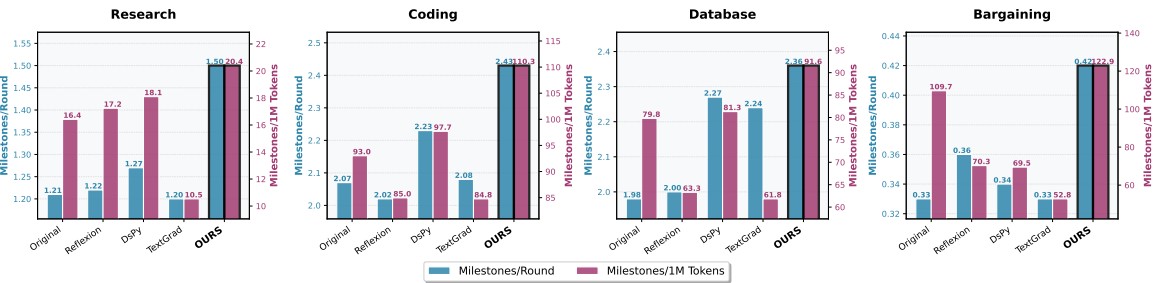

*Figure 3.* Execution efficiency measured by milestone achievement on *MultiAgentBench* per interaction round and per million tokens.

| | HE | HE-ET | MBPP | MBPP-ET | Average |
|---|---|---|---|---|---|
| | gpt-4o-mini + gpt-4o | | | | |
| Original | 94.40 | 81.20 | 87.50 | 57.80 | 80.23 |
| Reflexion | 93.10 | 79.90 | 87.00 | 57.00 | 79.25 |
| DsPy (MIPROv2) | 94.40 | 81.20 | 87.30 | 56.50 | 79.85 |
| TextGrad | 92.40 | 77.10 | 87.00 | 55.40 | 77.98 |
| TRUCE | **94.40** | **82.60** | **88.10** | **58.10** | **80.80** |
| | qwen-3-32b + qwen-3-235b | | | | |
| Original | 95.10 | 83.30 | **92.30** | 59.40 | 82.53 |
| Reflexion | 95.10 | 84.00 | 91.50 | 57.30 | 81.98 |
| DsPy (MIPROv2) | 95.10 | 81.20 | 90.20 | 58.40 | 81.23 |
| TextGrad | 95.80 | 81.90 | 91.20 | 59.40 | 82.08 |
| TRUCE | **96.50** | **86.80** | 91.00 | **59.70** | **83.50** |

*Table 3.* Pass@1 results on four programming datasets: *HumanEval* (*HE*), *HumanEval-ET* (*HE-ET*), *MBPP* and *MBPP-ET*.

| | Research | | Coding | |
| | TS | CS | TS | CS |
|---|---|---|---|---|
| MAPRO | 80.80 | 85.00 | 46.66 | 61.33 |
| HiveMind | 82.64 | 80.44 | 51.66 | 63.46 |
| TRUCE | **83.00** | **88.50** | **52.00** | **85.54** |

*Table 4.* Comparison with multi-agent prompt optimization baselines on representative collaborative domains under GPT-4o-mini.

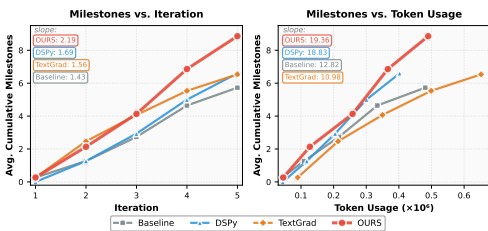

*Figure 4.* Accumulated milestone analysis on *Research* of *MultiAgentBench* with the number of interaction rounds (left) and with token usage in millions (right).

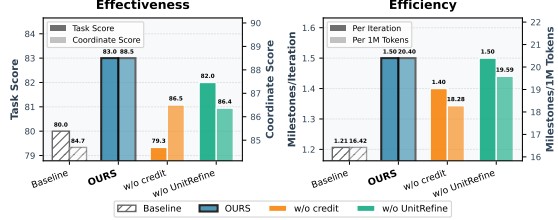

*Figure 5.* Ablation study comparing the full model (TRUCE) with variants that remove credit assignment (w/o Credit) or unit-based refinement (w/o UnitRefine).

optimization but consistently underperforms the full model. Although detailed feedback is available, applying it through

global prompt modifications proves unstable. Qualitative analysis (Figure 6) shows that credit assignment enables finer-grained policy-editing suggestions, while unit-based

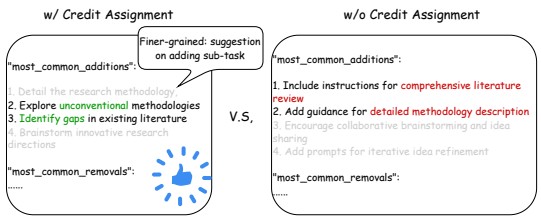

*(a)* Policy-Editing Suggestion Comparison

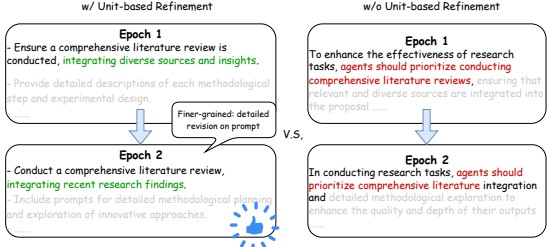

*(b)* Prompt Refinement Comparison.

*Figure 6.* Case study on the ablation study in the *Research* domain.

| Removal Strategy | TS | CS |
|---|---|---|
| TRUCE | 83.00 | 88.50 |
| Top-attributed | 81.45 ↓ | 80.83 ↓ |
| Random | 81.33 ↓ | 84.99 ↓ |
| Low/Mid-attributed | 83.33 ↑ | 83.85 ↓ |

*Table 5.* Attribution validation via sub-trajectory intervention on the Research domain.

refinement provides a stable interface for translating trajectory signals into effective prompt updates. Together, these results confirm that both components are necessary.

We further validate the quality of trajectory-level attribution through an intervention analysis on the *Research* domain. Specifically, we remove different subsets of trajectory segments during credit assignment and re-evaluate optimization outcomes. Removing top-attributed sub-trajectories causes the largest degradation (TS: $83.00 \rightarrow 81.45$, CS: $88.50 \rightarrow 80.83$), compared to random removal (TS: $83.00 \rightarrow 81.33$, CS: $88.50 \rightarrow 84.99$) or removing low-/mid-attributed segments (TS: $83.00 \rightarrow 83.33$, CS: $88.50 \rightarrow 83.85$). This supports that the attributed sub-trajectories capture behaviorally important evidence rather than arbitrary trajectory segments.

## 7. Conclusion

We investigated prompt optimization for LLM-based multi-agent systems under the practical constraint that agent roles and interaction structures are fixed, where prompts function as implicit behavioral policies. We identified trajectory-based credit assignment as the key challenge in this setting and proposed TRUCE, a trajectory-based framework that attributes outcome feedback to informative sub-trajectories and translates these signals into localized, unit-level prompt refinements. Experiments on MultiAgentBench and a state-of-the-art multi-agent programming benchmark show that TRUCE consistently improves task performance, coordination quality, and execution efficiency over strong baselines, while ablation studies confirm the necessity of both credit assignment and unit-based refinement. These results demonstrate that interpretable, trajectory-based prompt optimization enables stable and scalable improvement of multi-agent systems without modifying their underlying structure, providing a principled foundation for future work on automated prompt refinement in complex agentic environments.

## Acknowledgements

Bin Wu is supported by a Bloomberg Data Science Ph.D. Fellowship. We thank all the reviewers for their feedback. This work was partially supported by the New Generation Artificial Intelligence-National Science and Technology Major Project (2025ZD0122803), National Natural Science Foundation of China (52541003), 'Pioneer' and 'Leading Goose' R&D Program of Zhejiang (2026C01021, 2025C01129).

## Impact Statement

This paper presents work whose goal is to advance the field of Machine Learning. There are many potential societal consequences of our work, none which we feel must be specifically highlighted here.

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

# A. More Details about Methodology

We provide the pseudocode of TRUCE in Algorithm 1.

---

**Algorithm 1** Trajectory-based Rule Credit Estimation (TRUCE)

---

**Require:** Task distribution $\mathcal{D}$; multi-agent system $\mathrm{MAS}(\cdot; \{P_i\})$ with fixed roles and structure; initial prompts $\{P_i^{(0)}\}$ represented as policy units; evaluator $R(\mathcal{T}, r)$; number of optimization rounds $K$; aggregation parameter $k$

**Ensure:** Optimized prompts $\{P_i^{(K)}\}$

1: Initialize prompts $\{P_i^{(0)}\}$
2: **for** $t = 1$ to $K$ **do**
3:     Sample tasks $\{\mathcal{T}_j\} \sim \mathcal{D}$
4:     Initialize empty suggestion set $\mathcal{S}$
5:     **for all** tasks $\mathcal{T}_j$ **do**
6:         $(r_j, \mathcal{H}_j) \leftarrow \mathrm{MAS}(\mathcal{T}_j; \{P_i^{(t-1)}\})$
7:         $R_j \leftarrow R(\mathcal{T}_j, r_j)$
8:         Partition $\mathcal{H}_j$ into sub-trajectories $\{h_j^{(m)}\}$
9:         **for all** sub-trajectories $h_j^{(m)}$ **do**
10:             $c_j^{(m)} \leftarrow \Phi(h_j^{(m)}, R_j)$
11:         **end for**
12:         $\Delta_j \leftarrow \Psi(\{c_j^{(m)}\}, \{P_i^{(t-1)}\})$
13:         $\mathcal{S} \leftarrow \mathcal{S} \cup \Delta_j$
14:     **end for**
15:     $\mathcal{S}_{\mathrm{final}} \leftarrow \mathrm{TOPK}(\mathcal{S}, k)$
16:     $\{P_i^{(t)}\} \leftarrow \mathrm{APPLYEDITS}(\{P_i^{(t-1)}\}, \mathcal{S}_{\mathrm{final}})$
17: **end for**
18: **return** $\{P_i^{(K)}\}$

---

# B. More Details about Experimental Setup

## B.1. Benchmark

We evaluate our proposed TRUCE against competitive baselines on two benchmarks: *MultiAgentBench* (Zhu et al., 2025) and the *Programming* benchmark (Islam et al., 2025).

### B.1.1. *MultiAgentBench* BENCHMARK

*MultiAgentBench* comprises four domains with distinct interaction characteristics. The *Research* domain involves multi-stage collaborative workflows, including literature review, brainstorming, synthesis, and a 5Q-style research proposal. The *Coding* domain focuses on collaborative code generation and repair. The *Database* domain emphasizes structured querying, read–write operations, and consistency reasoning. The *Bargaining* domain evaluates multi-round negotiation, where agents must maintain strategic coherence and reach valid agreements under partially misaligned objectives.

For evaluation, we follow the original benchmark protocol and report three metrics: **Task Score**, **Coordination Score**, and **Milestones**. **Task Score** measures the quality of the final output. For the *Research* and *Bargaining* domains, scores are generated using an LLM-based rubric, while rule-based metrics are used for the remaining domains. **Coordination Score** quantifies agents' communication effectiveness and planning quality during interaction. **Milestones** provide a finer-grained measure of execution efficiency: each task is decomposed into a sequence of flexible intermediate milestones, and an LLM-based detector continuously monitors the interaction process to identify milestone completion. Following prior work, **Task Score** and **Coordination Score** serve as the primary evaluation metrics, while **Milestones** are used for efficiency analysis.

For each domain, we construct disjoint train/validation/test splits at the task-instance level, where each instance corresponds to a single benchmark task together with its full multi-agent execution episode (i.e., one input task, the resulting interaction

trajectory, and the evaluated final output). Specifically, we use 5/5/20 splits for *Research* and *Bargaining*, 5/5/15 for *Coding*, and 3/3/10 for *Database*, reflecting domain-specific task availability.

### B.1.2. *Programming* BENCHMARK

We additionally evaluate TRUCE on the *Programming* benchmark using a state-of-the-art multi-agent coding system. This setting is particularly challenging and practically relevant, as the underlying multi-agent system is carefully engineered by domain experts, leaving limited room for naive prompt optimization.

Following prior work (Islam et al., 2024; 2025), we consider four widely used programming datasets: *HumanEval* (Chen et al., 2021), *HumanEval-ET* (Dong et al., 2025), *MBPP* (Austin et al., 2021), and *MBPP-ET* (Dong et al., 2025). HumanEval-ET and MBPP-ET extend their respective base datasets by incorporating additional test cases. The problem set sizes of HumanEval (and HumanEval-ET) and MBPP (and MBPP-ET) are 164 and 397, respectively. We report the standard Pass@1 metric, where a task is considered successful if the single generated solution passes all test cases.

As the underlying multi-agent system, we adopt CODESIM (Islam et al., 2025), a recent state-of-the-art framework for collaborative program synthesis. CODESIM addresses planning, coding, and debugging through a human-inspired workflow and features explicit plan verification and internal debugging via step-by-step simulation of input–output behavior, enabling robust performance across programming tasks. We use the designed prompt from the original paper.

## B.2. Baselines

We compare our proposed TRUCE with the following baselines:

1. Original: we use the original prompts provided by the benchmark as the baselines for comparison. Note that the prompt here is without any optimization.

2. Reflexion (Shinn et al., 2023): this is a classic method, which takes the verbalized feedback as part of the prompt to construct the optimized prompt.

3. DsPy (Khattab et al., 2024): this is the evolution-based method that utilizes the numeric feedback to filter the better prompts. We follow their strong optimizer, MIPROv2 (Opsahl-Ong et al., 2024).

4. TextGrad (Yuksekgonul et al., 2025): this is one of the recent methods that uses text gradient as a proxy to differentiate from the verbalized feedback, which is used for refining the prompt.

5. MAPRO (Zhang et al., 2026): a recent multi-agent prompt optimization method that formulates prompt refinement as maximum a posteriori inference over agent prompts.

6. HiveMind (Xia et al., 2026): a recent multi-agent prompt optimization method that performs contribution-guided online prompt optimization for LLM multi-agent systems.

## B.3. Prompts

We list the used prompts in our experiments.

---

**Prompt for Trajectory-Based Credit Assignment**

```
Please evaluate the quality and progress of the current iteration in this
    multi-agent collaboration.

Current Iteration Context:
Iteration Number: {iteration_number}
Iteration Content: {iteration_content}

Task Context:
Task Background: {task_context}
Previous Iterations: {previous_iterations}
```

---

```
Evaluation of final result:
{global_evaluation}

evaluation_criteria:

iteration_progress:
description: "Progress made in current iteration"
evaluation_points:
"Whether meaningful progress was made"
"Whether intermediate results are valuable"
"Whether the direction is correct and effective"

agent_coordination:
description: "Agent coordination in current iteration"
evaluation_points:
"Whether agents worked together effectively"
"Whether communication was clear and purposeful"
"Whether work division was appropriate"

stage_appropriateness:
description: "Evaluate whether the output meets the current stage"
evaluation_points:
"Whether the content depth is suitable for the current stage"
"Whether it is prepared for the subsequent work"
"Whether the time arrangement is reasonable"

output_format:
Please return the evaluation results in JSON format:
{
"iteration_progress": {
"score": score(1-10),
"analysis": "progress analysis",
"key_achievements": ["key achievement 1", "key achievement 2"]
},
"agent_coordination": {
"score": score(1-10),
"analysis": "coordination analysis",
"coordination_highlights": ["coordination highlight 1", "coordination highlight 2"],
"coordination_issues": ["coordination issue 1", "coordination issue 2"]
},
"stage_appropriateness": {
"score": score(1-10),
"analysis": "stage appropriateness analysis",
"alignment_evidence": ["alignment evidence 1", "alignment evidence 2"]
},
"iteration_summary": {
"overall_score": average score,
"main_contributions": ["main contribution 1", "main contribution 2"],
"areas_for_improvement": ["area for improvement 1", "area for improvement 2"],
"next_iteration_suggestions": ["next iteration suggestion 1", "next iteration
    suggestion 2"]
}
}
```

---

**Prompt for Generating Policy-Editing Suggestions**

```
 Please analyze the following Agent's performance and provide improvement
    recommendations based on final result quality:

**Task Context:**
{task_context}

**Final Result Quality Assessment:**
{result_context}

**Full Result Evaluation:**
{result_evaluation}

**Agent Information:**
- Agent ID: {agent_history['agent_id']}
- Agent Type: {agent_history['agent_type']}
- Profile: {agent_history['profile']}

**Current System Prompt:**
{agent_history['original_system_prompt']}

**Agent Historical Performance:**
- Number of tasks executed: {len(agent_history['tasks_performed'])}
- Number of communications: {len(agent_history['communications'])}
- Token usage: {agent_history['token_usage']}

**Detailed Task History:**
{agent_history['tasks_performed']}...

**Communication History:**
{agent_history['communications']}...

**Result History:**
{agent_history['results']}...

**Result Quality-Based Deep Analysis Requirements:**

1. **Result-Oriented Assessment**: Analyze whether this agent's contributions are
    effective based on final result quality
2. **Causality Analysis**: Analyze the causal relationship between this agent's
    behavior and final result quality
3. **Impact Verification**: Whether this agent's work had positive impact on final
    results
4. **Problem Attribution**: If result quality is poor, whether this agent is one of
    the influencing factors
5. **Targeted Improvement**: Provide targeted improvement recommendations based on
    result quality issues

**Important**: Please judge agent performance effectiveness by combining result
    quality assessment, not just process analysis.

Please return evaluation results in JSON format:
{{
    "overall_score": 1-10,
    "result_oriented_analysis": {{
        "contribution_to_final_result": "Analysis of this agent's specific
    contribution to final result",
        "effectiveness_rating": 1-10,
        "impact_on_quality": "Analysis of impact on result quality"
    }},
    "strengths": ["Strength 1 based on result verification", "Strength 2 based on
    result verification", ...],
```

```
    "weaknesses": ["Weakness 1 affecting result quality", "Weakness 2 affecting
result quality", ...],
    "causality_analysis": {{
        "positive_contributions": ["Behavior 1 promoting result quality", "Behavior
2 promoting result quality"],
        "negative_impacts": ["Behavior 1 damaging result quality", "Behavior 2
damaging result quality"],
        "missed_opportunities": ["Missed opportunity 1 to improve results", "Missed
opportunity 2 to improve results"]
    }},
    "prompt_suggestions": {{
        "result_oriented_improvements": "Prompt improvement suggestions based on
result quality issues",
        "effectiveness_enhancements": "Prompt modifications to improve actual
effectiveness",
        "quality_focus_additions": "Prompt additions to enhance result quality
awareness",
        "collaboration_optimizations": "Prompt adjustments to optimize
collaboration effects"
    }},
    "targeted_recommendations": {{
        "immediate_fixes": ["Immediate improvement 1", "Immediate improvement 2"],
        "strategic_improvements": ["Strategic improvement 1", "Strategic
improvement 2"],
        "quality_assurance_measures": ["Quality assurance measure 1", "Quality
assurance measure 2"]
    }},
    "specific_prompt_modifications": {{
        "add_instructions": ["Specific instruction 1 to add", "Specific instruction
2 to add"],
        "remove_content": ["Content 1 to remove", "Content 2 to remove"],
        "restructure_suggestions": ["The content to be modified 1(and the way to
modify)", "The content to be modified 2(and the way to modify)"]
    }}
}}
```

### Prompt for Suggestion-Aggregated Policy Refinement

```
 You are given a list of short rules/suggestions (may be redundant or semantically
    similar).
Task:
(1) normalize/merge near-duplicates;
(2) rank by (estimated frequency + practical importance);
(3) return the top-{top_k} representative and concise items.

Input items (one per line with index):
{input_items}

Return STRICT JSON only:
{{
  "top": [
    {{
      "text": "representative concise rule",
      "support_examples_idx": [1,5,9],
      "support_count": 3,
      "importance": 0
    }}
```

```
            ]
    }}
```

## B.4. Optimization Budget and Complexity Analysis

**Data Efficiency**  A key advantage of TRUCE is its sample efficiency. Unlike black-box optimizers that may require large datasets to estimate gradients, TRUCE extracts dense, unit-level supervision from individual trajectories. Consequently, we utilize small, high-quality training sets for optimization: 5 training task instances for Research/Coding/Bargaining and 3 for Database (see Appendix B.1.1).

**Optimization Overhead**  Prompt optimization represents a one-time *offline* investment, distinct from the recurring *online* cost of inference (Operating Expenditure). While TRUCE incurs a marginal overhead for trajectory analysis during this offline phase, it yields agents that are significantly more efficient during deployment. As demonstrated in Figure 3, TRUCE-optimized agents achieve task milestones with substantially fewer interaction rounds and tokens compared to baselines. In practical deployments, the recurring savings from this enhanced inference efficiency quickly amortize the initial optimization cost, rendering the offline overhead an acceptable trade-off for long-term scalability.

## C. Theory

This appendix provides a formal abstraction and analysis of the optimization behavior of TRUCE, complementing the intuition in Section 4.6 of the main paper. Our goal is not to model the full complexity of LLM-based multi-agent execution, but to capture how trajectory-based credit assignment and unit-level aggregation restrict the effective prompt-editing space and lead to stable local improvement.

Let $\mathcal{P}$ be a finite set of candidate prompts, each represented as a collection of unit-level behavioral rules, and closed under the unit-based editing operations used by TRUCE. Each prompt $P \in \mathcal{P}$ induces a stochastic multi-agent system rollout on a task $\mathcal{T}$ sampled from a distribution $D$, producing a bounded scalar reward $R(P, \mathcal{T}) \in [0, 1]$ that abstracts the task-level evaluation metric used in practice. Define the objective

$$J(P) := \mathbb{E}_{\mathcal{T} \sim D}[R(P, \mathcal{T})].$$

Denote $\mathcal{N}_{\text{all}}(P)$ as the set of prompts that differ from $P$ by a single unit-level edit. Trajectory-based credit assignment induces an effective neighborhood $\mathcal{N}(P) \subseteq \mathcal{N}_{\text{all}}(P)$ by restricting attention to unit-level edits that are repeatedly suggested based on execution evidence. Given $\varepsilon > 0$, we say a prompt $P$ is $\varepsilon$-locally optimal if

$$J(P) \geq \max_{Q \in \mathcal{N}(P)} J(Q) - \varepsilon.$$

An abstracted optimization procedure capturing the behavior of TRUCE can be summarized as follows: at each iteration $k$, given current prompt $P_k$:

- For each candidate $Q \in \mathcal{N}(P_k) \cup \{P_k\}$, estimate the objective $J(Q)$ by Monte Carlo:

$$\hat{J}_k(Q) = \frac{1}{n} \sum_{i=1}^{n} R(Q, \mathcal{T}_i), \quad \mathcal{T}_i \overset{i.i.d.}{\sim} D.$$

- Let $Q_k^\star := \arg\max_{Q \in \mathcal{N}(P_k)} \hat{J}_k(Q)$.

- Accept the move $P_{k+1} = Q_k^\star$ only if it clears a margin: $\hat{J}_k(Q_k^\star) \geq \hat{J}_k(P_k) + 2\beta$, where $\beta = \sqrt{\frac{\log(2M/\delta)}{2n}}$ and $M := \max_P |\mathcal{N}(P)| + 1$ denotes the maximum size of all neighborhoods. Otherwise stop and output $P_k$.

**Theorem C.1** (High-probability convergence to an $\varepsilon$-local optimum). *Fix $\varepsilon > 0$ and confidence level $\delta \in (0, 1)$. Choose $n \geq \frac{2}{\varepsilon^2} \log\left(\frac{2MK}{\delta}\right)$, where $K$ is an upper bound on the number of iterations and $M$ is the maximum size of all neighborhoods. Then with probability at least $1 - \delta$:*

1. *(No false improvements) Every accepted move satisfies:* $J(P_{k+1}) \geq J(P_k) + \varepsilon$.

2. *(Local optimality at stop) When the algorithm stops at $\widehat{P}$, it is an $\varepsilon$-local optimum:* $J(\widehat{P}) \geq \max_{Q \in \mathcal{N}(\widehat{P})} J(Q) - \varepsilon$.

*Proof.* Fix $k \in \{0, \ldots, K-1\}$ and a candidate $Q \in \mathcal{N}(P_k) \cup \{P_k\}$. By Hoeffding's inequality,

$$\Pr\left(\left|\widehat{J}_k(Q) - J(Q)\right| > \tfrac{\varepsilon}{2}\right) \ \leq \ 2\exp\left(-2n \cdot (\tfrac{\varepsilon}{2})^2\right) \ = \ 2\exp\left(-\frac{n\varepsilon^2}{2}\right).$$

Define the probabilistic event

$$\mathcal{E} \ := \ \bigcap_{k=0}^{K-1} \ \bigcap_{Q \in \mathcal{N}(P_k) \cup \{P_k\}} \left\{\left|\widehat{J}_k(Q) - J(Q)\right| \leq \tfrac{\varepsilon}{2}\right\}.$$

By a union bound over at most $MK$ empirical estimates,

$$\Pr(\mathcal{E}^c) \ \leq \ MK \cdot 2\exp\left(-\frac{n\varepsilon^2}{2}\right).$$

Under the stated choice $n \geq \frac{2}{\varepsilon^2} \log\left(\frac{2MK}{\delta}\right)$ in the theorem, we have $2MK \exp(-n\varepsilon^2/2) \leq \delta$, hence
$$\Pr(\mathcal{E}) \ \geq \ 1 - \delta.$$

In the remainder, condition on $\mathcal{E}$. Suppose the algorithm accepts a move from $P_k$ to $P_{k+1}$. By the acceptance rule (implicit in the theorem statement), acceptance implies $\widehat{J}_k(P_{k+1}) \geq \widehat{J}_k(P_k) + \varepsilon$. On $\mathcal{E}$ we have $\widehat{J}_k(P_{k+1}) \leq J(P_{k+1}) + \varepsilon/2$ and $\widehat{J}_k(P_k) \geq J(P_k) - \varepsilon/2$, so

$$J(P_{k+1}) \ \geq \ \widehat{J}_k(P_{k+1}) - \tfrac{\varepsilon}{2} \ \geq \ \widehat{J}_k(P_k) + \varepsilon - \tfrac{\varepsilon}{2} \ \geq \ J(P_k) - \tfrac{\varepsilon}{2} + \varepsilon - \tfrac{\varepsilon}{2} \ = \ J(P_k) + \varepsilon,$$

which proves the first claim.

Assume the algorithm stops at $\widehat{P}$ after some iterations. Stopping means that no neighbor clears the $\varepsilon$ improvement threshold, i.e. for all $Q \in \mathcal{N}(\widehat{P})$, $\widehat{J}_k(Q) < \widehat{J}_k(\widehat{P}) + \varepsilon$. On $\mathcal{E}$, $\widehat{J}_k(Q) \geq J(Q) - \varepsilon/2$ and $\widehat{J}_k(\widehat{P}) \leq J(\widehat{P}) + \varepsilon/2$, hence for every neighbor $Q$,

$$J(Q) - \tfrac{\varepsilon}{2} \ \leq \ \widehat{J}_k(Q) \ < \ \widehat{J}_k(\widehat{P}) + \varepsilon \ \leq \ J(\widehat{P}) + \tfrac{\varepsilon}{2} + \varepsilon \ = \ J(\widehat{P}) + \tfrac{3\varepsilon}{2}.$$

Therefore $J(Q) \leq J(\widehat{P}) + 2\varepsilon$ for all $Q \in \mathcal{N}(\widehat{P})$, i.e.

$$J(\widehat{P}) \ \geq \ \max_{Q \in \mathcal{N}(\widehat{P})} J(Q) - 2\varepsilon,$$

so $\widehat{P}$ is a $2\varepsilon$-local optimum. Replacing the above $\varepsilon$ with $\varepsilon/2$ concludes the proof of the second claim. $\square$

Theorem C.1 provides a high-probability guarantee for a stylized abstraction of TRUCE. In this abstraction, candidate prompt edits are evaluated via explicit Monte Carlo estimates of the expected task objective. In TRUCE, Monte Carlo sampling is realized at the task level by repeatedly executing the multi-agent system on tasks sampled from $\mathcal{D}$. Rather than explicitly re-evaluating each edited prompt, candidate unit-level edits are assessed implicitly through trajectory-based credit signals and their frequency across sampled tasks, with aggregation serving as an evidence-thresholded acceptance mechanism. Explicitly evaluating the expected performance of every candidate unit-level edit would require re-running the full multi-agent system for each neighbor, which is impractical in long-horizon, tightly coupled multi-agent settings. The theorem thus formalizes the role of unit-level refinement and aggregation in restricting the effective edit space, reducing variance, and preventing spurious prompt updates.

# D. More Analysis

## D.1. Additional Analysis on MultiAgentBench

We observe systematic differences between task score (TS) and coordination score (CS) improvements across domains. In collaborative settings, CS exhibits larger relative gains than TS, particularly in the Coding and Database domains, indicating improved communication efficiency, role adherence, and planning. These results suggest that trajectory-based credit assignment primarily enhances interaction dynamics, which then translate into downstream task-level improvements.

| Method | Research | | Bargaining | |
|---|---|---|---|---|
| | TS | CS | TS | CS |
| Original | $80.40 \pm 0.25$ | $86.20 \pm 0.40$ | $73.52 \pm 0.71$ | $82.83 \pm 0.33$ |
| Reflexion | $80.33 \pm 0.63$ | $87.98 \pm 0.87$ | $79.17 \pm 0.53$ | $82.79 \pm 0.27$ |
| DsPy | $82.07 \pm 0.39$ | $85.41 \pm 0.89$ | $60.19 \pm 0.93$ | $79.35 \pm 0.21$ |
| TextGrad | $81.27 \pm 0.57$ | $88.03 \pm 0.78$ | $64.80 \pm 0.65$ | $80.68 \pm 0.07$ |
| TRUCE | $\mathbf{83.13 \pm 0.50}$ | $\mathbf{89.35 \pm 0.46}$ | $\mathbf{82.55 \pm 0.80}$ | $\mathbf{83.50 \pm 0.64}$ |

*Table 6.* Robustness across five runs on representative collaborative and competitive domains.

*Table 7.* Sensitivity to optimizer strength under the Research domain.

| Optimizer | Executor | TS | CS |
|---|---|---|---|
| None | GPT-4o-mini | 80.00 | 84.69 |
| GPT-4o-mini | GPT-4o-mini | 80.66 | **92.40** |
| GPT-4o | GPT-4o-mini | **83.00** | 88.50 |

### D.2. Robustness Analysis

To assess robustness under stochastic LLM execution, we repeat evaluation across five random seeds on representative collaborative (*Research*) and competitive (*Bargaining*) domains. Table 6 reports mean and standard deviation for task score (TS) and coordination score (CS). TRUCE consistently achieves the strongest performance with low variance across both settings, indicating that the observed gains are stable despite stochastic execution.

### D.3. Sensitivity to Optimizer Strength

To assess dependence on optimizer capability, we evaluate a self-evolving setting where GPT-4o-mini is used for both optimization and execution. Table 7 shows that TRUCE remains effective even when optimization is performed with the same weaker model used for execution, suggesting that improvements primarily come from the optimization framework rather than optimizer scale alone.

### D.4. Extended Efficiency Analysis

Beyond aggregate milestone counts, we analyze accumulated milestone curves over interaction rounds and token usage. The slope of these curves reflects the rate of task progress under a fixed execution budget. Across domains, our method consistently exhibits steeper slopes than all baselines, indicating faster progress throughout long-horizon execution.

### D.5. Optimization Cost

We additionally analyze optimization cost on the *Research* domain. A full optimization process for TRUCE requires 1,526 LLM calls and approximately 7.5M tokens. In comparison, DsPy requires substantially more resources (3,470 calls and 14.3M tokens) due to its search-heavy optimization process, while TextGrad is more lightweight (628 calls and 2.2M tokens) because it applies global prompt updates without explicit trajectory decomposition or fine-grained credit assignment. Overall, TRUCE provides a practical balance between optimization cost and fine-grained multi-agent optimization capability.

### D.6. Qualitative Case Study for Ablation (Research)

Figure 6 provides a qualitative illustration of how trajectory-based credit assignment and unit-based prompt refinement contribute to TRUCE. We analyze the same ablation settings as in Section 6.4.

**Effect of credit assignment.** Figure 6(a) compares policy-editing suggestions generated with and without trajectory-based credit assignment. With credit assignment enabled, the suggestions are fine-grained and localized, targeting specific missing or erroneous subtasks (e.g., identifying gaps in the literature). In contrast, without credit assignment, the suggestions are

generic and coarse, offering high-level advice that does not pinpoint where or why failures occur. This qualitative difference explains the performance degradation observed when credit assignment is removed.

**Effect of verbalized policy units.** Figure 6(b) illustrates the role of unit-based, verbalized policy representations in prompt refinement. When prompts are represented as explicit policy units, refinements modify specific behavioral rules (e.g., emphasizing integration of recent research findings). Without verbalized policy units, updates are applied globally to the prompt, leading to broader but less precise changes. These global modifications are more likely to introduce unintended behavioral shifts, accounting for the instability observed in the corresponding ablation.

Together, these observations qualitatively support the quantitative ablation results in Section 6.4, demonstrating that both trajectory-based credit assignment and unit-level prompt representations are necessary for stable and effective prompt optimization.

