# OpenReview forum: "From Interaction Trajectories to Prompt Rules: Credit Assignment for Multi-Agent Prompt Optimization"
_ICML.cc/2026/Conference — ICML 2026 regular_

### Official Review · Reviewer_JGaW · 2026-03-09

**Soundness:** 3
**Presentation:** 3
**Significance:** 2
**Originality:** 2
**Overall Recommendation:** 4
**Confidence:** 2

**Summary:**

In this paper, the authors study an actively researched problem in prompt-optimization, i.e. credit assignment. The existing suite of prompt-optimization often struggle to identify how to convert signals from an evaluation into targeted feedback; a problem that is exacerbated in multi-agent settings. The authors adopt a four-stage process to perform trajectory-based prompt analysis. Step 1 - LLM-decomposition of the trajectory into sub-trajectories, Step 2 - inferring credit for each sub-trajectory, Step 3 - Generating prompt edits based on unit-based prompt representations and credit signals, Step 4 - Aggregate support for suggestions and select the top-k suggestions. The authors test their approach on the MultiAgentBench benchmark, and show that their approach outperforms Reflexion, DsPy, and TextGrad with regards to task-performance and coordination across most tasks.

**Compliance With Llm Reviewing Policy:**

Affirmed.

**Final Justification:**

The additional comparisons with multi-agent baselines and the inclusion of standard deviations improved my assessment of the paper.

**Key Questions For Authors:**

- My primary critique of this paper pertains to a lack of a multi-agent baseline. Can the authors justify this choice?
- Instead of a constrained suggestion setup, I’d be curious to see how the optimizer performed when the suggestions (delta) were open-ended. I recognize that open-ended suggestions make it more difficult to measure “support,” however, did the authors conduct any experiments along these lines to understand how open-ended suggestions perform within their pipeline?

**Limitations:**

The authors have not discussed any limitations of their work, nor potential negative societal impacts stemming from their work. Given the extensive reliance on LLMs within the optimization process, it would be valuable to discuss the potential for LLM drift and how it might lead to the propagation of harmful or unhelpful patterns in the generated outputs.

**Strengths And Weaknesses:**

## Strengths

- I appreciated the authors approach towards creating unit-based scaffolding around prompts to create a more intelligible means of prompt accountability.
- The authors show that both trajectory-based credit assignment and unit-based prompt breakdown are necessary for the optimal performance of their system.
- Given that this approach is very LLM-heavy, with multiple model-calls required for each task, the efficiency analysis provided by the authors was helpful to gauge the utility of their approach compared to other methods.

## Weaknesses

- The authors only compare their approach to weaker prompt-optimization methods designed to optimize single-agent prompts. I think a fairer evaluation would be to compare against existing multi-agent prompt optimization approaches, such as [1,2]. Unless the authors have a strong rationale for not including such a comparison, I think this is a significant weakness of the paper.
- This paper’s claim that they are the first to investigate credit-assignment through trajectory analysis is incorrect. There are recent works which have approached this credit assignment problem via topological analysis[1], or via shapley values[2].
- It would be helpful to include standard-errors (or some equivalent indicator of variance of the task/coordination scores) to better understand the significance of the differences in performance.

[1] - Zhang, Zheyuan, et al. "MAPRO: Recasting Multi-Agent Prompt Optimization as Maximum a Posteriori Inference." arXiv preprint arXiv:2510.07475 (2025).
[2] - Xia, Yihan, et al. "HiveMind: Contribution-Guided Online Prompt Optimization of LLM Multi-Agent Systems." arXiv preprint arXiv:2512.06432 (2025).

---

> ### Author Rebuttal · Authors · 2026-03-31
>
> We thank the reviewer for the constructive feedback and helpful suggestions.
>
> **(1) Multi-agent baselines.**
>
> We agree that including multi-agent prompt optimization baselines is important for fair comparison. The suggested works (MAPRO, EACL’26; HiveMind, AAAI’26) are very recent. We have now implemented both baselines following their descriptions and added them to our evaluation. Despite their strong performance, TRUCE consistently outperforms both methods on MultiAgentBench (e.g., +2.2 TS / +3.5 CS over MAPRO and +0.36 TS / +8.06 CS over HiveMind in the Research domain, and +5.34 TS / +24.21 CS over MAPRO and +0.34 TS / +22.08 CS over HiveMind in the Coding domain). This strengthens our empirical claim that trajectory-aware, unit-level refinement provides additional benefits beyond existing MAS optimization approaches.
>
>
> **(2) Variance and statistical robustness.**
>
> We have added multi-run statistics and statistical significance tests. On both a collaborative domain (Research) and a competitive domain (Bargaining), TRUCE consistently improves over baselines with low variance. For example, on Research, TRUCE achieves TS = 83.13 ± 0.50 vs. the best baseline (DSPy: 82.07 ± 0.39, p = 0.006) and CS = 89.35 ± 0.46 vs. TextGrad (88.03 ± 0.78, p = 0.015), both statistically significant. On Bargaining, TRUCE achieves TS = 82.55 ± 0.80 vs. Self-Refine (79.17 ± 0.53, p < 0.001), while CS also improves (83.50 vs. 82.83) but is not statistically significant (p = 0.085). Overall, these results confirm that the performance gains are stable and statistically meaningful across runs.
>
>
> **(3) Open-ended suggestions vs. constrained edits.**
>
> We appreciate this question and agree it highlights an important design choice. Our “w/o unit refine” ablation approximates a less constrained editing regime, where credit signals are not consistently mapped to localized policy units. We observe that this variant leads to clear performance degradation (Fig. 5–6), indicating that unconstrained updates are less effective. This is because open-ended edits introduce higher semantic variability and make it difficult to aggregate support across trajectories. In contrast, unit-level edits provide a structured update space that enables consistent credit propagation and stable aggregation. These results suggest that the constrained edit design in TRUCE is important for reliable optimization rather than a limitation of the method.
>
> **(4) Novelty clarification.**
>
> We appreciate the pointer to related work. We do not claim to be the first to study credit assignment in multi-agent prompt optimization; rather, as noted in the related work section, we position this as an open challenge. Prior works such as MAPRO and HiveMind also explore credit assignment mechanisms. Our contribution differs in two key aspects: (i) we use verbalized credit signals derived from trajectory analysis, rather than value-based or contribution-estimation formulations; and (ii) we propagate these signals to localized policy units within prompts, enabling fine-grained, interpretable edits, whereas prior methods typically operate at the agent-level or prompt-level. We will revise the wording to better clarify this distinction.
>
>
> **(5) Limitations and societal impact.**
>
> We will expand the discussion of limitations. In particular, TRUCE relies on LLM-based optimization, which may introduce drift or propagate suboptimal editing patterns. In addition, our current framework focuses on fixed agent roles and interaction structures, and extending it to more dynamic MAS settings is an important direction for future work.
>
> We hope these additional analyses and clarifications address the reviewer’s concerns.

---

> > ### Author Rebuttal · Reviewer_JGaW · 2026-04-08
> >
> > I thank the authors for their response. I have updated my score accordingly.

---

### Official Review · Reviewer_vmHa · 2026-03-11

**Soundness:** 2
**Presentation:** 3
**Significance:** 3
**Originality:** 2
**Overall Recommendation:** 4
**Confidence:** 3

**Summary:**

This paper proposes TRUCE, a prompt-optimization framework for LLM-based multi-agent systems with fixed agent roles and interaction structures. The key idea is to link outcome feedback to informative sub-trajectories for credit attribution and translate the resulting signals into localized prompt edits over behavioral rules.

**Compliance With Llm Reviewing Policy:**

Affirmed.

**Final Justification:**

The paper is clear and reasonably motivated, but I remain unconvinced about its soundness, methodological justification, and overall significance relative to its claims. The rebuttal improved clarity and added useful evidence, but it did not materially change my evaluation, so I keep my original score unchanged and view the paper as remaining at the borderline level.

**Key Questions For Authors:**

1. How are the policy units constructed in practice, and what criteria are used to define or segment them from the original prompts?

2. Can the authors report results across multiple random seed runs, with confidence intervals?

3. The experiments use a stronger optimizer model to refine a smaller task model. How much of the gain is due to TRUCE itself rather than this setup?

**Limitations:**

TRUCE relies heavily on prompted LLM judgments throughout the optimization pipeline, so its effectiveness may depend on the quality and stability of the underlying optimizer model. In addition, the experiments are conducted on relatively small instance-level splits, which makes it difficult to assess robustness and variance more broadly.   Finally, because the framework is designed for fixed agent roles and interaction structures, it is still unclear how well it would extend to more dynamic multi-agent settings.

**Strengths And Weaknesses:**

Strengths:

1. The paper is well written and easy to follow, with strong motivation for the problem setting.

2. The paper studies an important and timely problem and proposes a simple yet effective method.

3. The empirical results are encouraging across multiple domains, and the ablation study provides some support for the usefulness of trajectory-level attribution and unit-based refinement.

Weaknesses:

1. The core components of the method, including trajectory analysis, credit attribution, policy-editing suggestion generation, and aggregation, are all implemented via prompted LLM judgments. As a result, the method relies heavily on heuristic LLM reasoning rather than on a more clearly specified, verifiable, or explicitly learned attribution mechanism.

2. The method description is relatively brief, with several key implementation details left unclear, such as how policy units are defined and how trajectory-level evidence is mapped to specific edits.

3. The terminology around “instances” is not sufficiently precise. Appendix B.4 mentions “5 training instances” and “3 instances”, but the paper does not explicitly define whether an instance denotes a task example, an episode, or another benchmark unit. Clarifying this would make the experimental setup and the associated data-efficiency claim easier to interpret.

4. Because the method relies heavily on prompted LLM judgments and appears to optimize on very small instance-level splits, additional robustness reporting (e.g., variance estimates, confidence intervals, or multi-run results) would strengthen the empirical claims.

5. The experimental section is somewhat under-described. Many results are presented mainly in tables with limited detailed analysis. More implementation detail and deeper discussion would improve reproducibility and make the empirical claims easier to assess.

6. Several implementation details of the method are missing or unclear. Providing the source code would therefore be valuable for validating the reproducibility of the work.

---

> ### Author Rebuttal · Authors · 2026-03-31
>
> We thank the reviewer for the positive assessment and helpful feedback. We provide additional clarifications and robustness analysis to address the concerns.
>
> **(1) Policy unit construction.**
>
> Policy units are not fixed manual annotations, nor are they extracted by parsing the original benchmark prompts. Instead, the original prompt serves as a fixed foundational context (e.g., base agent roles). Policy units are automatically constructed and updated during optimization as semantic, rule-like behavioral components. When trajectory feedback identifies a missing behavior, TRUCE adds a new discrete unit. We prioritize (i) behavioral independence, so each unit captures a distinct aspect of agent behavior, and (ii) actionability, ensuring each can be directly edited. Consequently, a unit introduced in one round may be refined or deleted in subsequent rounds based on new trajectory evidence. Any conflicting or overlapping edits are safely reconciled during the cross-task aggregation stage before updates are applied.
>
>
> **(2) Variance and robustness.**
>
> We have added multi-run statistics across different random seeds. On MultiAgentBench, TRUCE achieves consistent improvements with low variance across both collaborative (Research) and competitive (Bargaining) domains. For example, on Research, TRUCE obtains a Task Score of 83.13 ± 0.50 (compared to DSPy's 82.07 ± 0.39), while on Bargaining it achieves 82.55 ± 0.80 (compared to Self-Refine's 79.17 ± 0.53). Similar stable trends hold for Coordination Scores. These tight variance bounds indicate that the optimization is robust and stable, even with stochastic LLM components and small training splits.
>
> **(3) Effect of optimizer strength.**
>
> To isolate the effect of using a stronger optimizer, we evaluate a self-evolving setting where the same model (GPT-4o-mini) is used for both optimization and execution. In this setting, TRUCE still improves over the non-optimized baseline (TS: 80.00 → 80.66; CS: 84.69 → 92.40), indicating that the gains are not solely due to using a stronger optimizer. Using a stronger optimizer (GPT-4o) further improves TS (80.66 → 83.00), while CS remains strong (92.40 → 88.50). This suggests that optimizer strength primarily affects the quality and direction of updates (e.g., emphasizing task completion vs. coordination), rather than being the sole driver of performance gains. Overall, these results confirm that the improvements arise from the trajectory-aware, unit-level optimization process itself.
>
> **(4) On LLM-based components.**
>
> We acknowledge that TRUCE relies on prompted LLM modules for trajectory analysis and edit generation. However, rather than a simple heuristic wrapper, the framework defines a formal local optimization algorithm over a discrete space of policy units. LLMs act as the proposal and scoring functions within this algorithm, restricting the search space to evidence-backed updates. This modular design means the LLM components could be seamlessly swapped for learned or rule-based attribution mechanisms in future work.
>
> **(5) Clarification of “instance”.**
>
> An “instance” refers to a single task episode (i.e., one full multi-agent interaction trajectory for a given task input). Training instances are used for optimization, while evaluation is performed on separate test instances.
>
> **(6) Reproducibility and experimental details.**
>
> We will expand implementation details in the final version and release code to facilitate reproducibility.
>
>
> We hope these clarifications and additional results address the reviewer’s concerns.

---

> > ### Author Rebuttal · Reviewer_vmHa · 2026-04-01
> >
> > The rebuttal improves clarity and adds useful evidence, but it does not materially change my assessment of the paper’s soundness and methodological clarity. I therefore keep my original score unchanged. In my view, the paper remains at the borderline level.

---

### Official Review · Reviewer_REfD · 2026-03-12

**Soundness:** 2
**Presentation:** 2
**Significance:** 2
**Originality:** 2
**Overall Recommendation:** 2
**Confidence:** 3

**Summary:**

This paper introduces TRUCE, a trajectory-based prompt optimization framework for LLM-based multi-agent systems. The key idea is to address the credit assignment problem by attributing outcome feedback to informative sub-trajectories and translating these signals into localized prompt edits over interpretable policy units. Prompts are decomposed into rule-like units that can be modified, and suggested edits are aggregated across tasks to stabilize updates. Experiments on MultiAgentBench and a multi-agent programming benchmark show consistent improvements over existing baselines.

**Compliance With Llm Reviewing Policy:**

Affirmed.

**Final Justification:**

I read the rebuttal and follow-up clarification by the authors, but I am keeping my score unchanged. The rebuttal improves the paper and adds useful evidence, especially the additional intervention-style analyses, multi-run statistics, and optimization-cost discussion.   However, my central concerns remain only partially resolved.
Most importantly, the new masking experiment is a meaningful step in the right direction, but it does not fully resolve my concern about whether TRUCE can reliably attribute specific behavioral evidence in sub-trajectories to specific prompt-rule edits, rather than merely identifying generally important trajectory segments.
In addition, the method still appears to rely heavily on prompted LLM judgments at nearly every core step—attribution, edit generation, reconciliation, and refinement—so I remain unconvinced that this is a well-specified optimization algorithm in a strong sense rather than a structured LLM-driven pipeline.
The additional clarifications on sub-trajectory segmentation and policy units are helpful, but they are still not detailed enough to fully assess reproducibility or sensitivity to these design choices across different MAS protocols.
Overall, I appreciate the thoughtful rebuttal and the additional analyses, but I do not think they fully resolve the core issues that shaped my original assessment.

**Key Questions For Authors:**

1. How are sub-trajectories defined in practice across different MAS protocols?

2. How are policy units initially constructed from the original prompts? Are they manually defined, automatically extracted, or generated by an LLM? Additionally, how are conflicts handled when multiple edits affect overlapping units or when edits modify previously added rules?

3. Could the authors report the optimization-time overhead of TRUCE (e.g., number of additional LLM calls, token usage, or wall-clock time per optimization round) and compare it to baselines such as TextGrad or DSPy?

**Limitations:**

yes

**Strengths And Weaknesses:**

### Strengths

1. The paper focuses on prompt optimization in multi-agent systems where the underlying LLMs, agent roles, and interaction structures are fixed. This setting is highly relevant in real-world deployments involving frozen or proprietary models, where prompt refinement is often the primary mechanism for improving system behavior.

2. Representing prompts as interpretable policy units and editing them through structured operations provides a clean interface for targeted updates. This design supports minimal and interpretable changes and aligns well with how practitioners iteratively refine prompts in multi-agent systems.


### Weaknesses

1. **Credit assignment quality is not directly validated.**
   The central claim of the paper is that trajectory-level evidence can be attributed to specific prompt rules. However, the paper does not provide a direct evaluation of attribution quality. The proposed credit signals are verbal and correlational, and there is no counterfactual or intervention-based validation showing that the identified sub-trajectories or policy units are causally responsible for the final outcome.

2. **The approach relies heavily on LLM-driven heuristics.**
   Key components—including trajectory attribution, edit generation, suggestion reconciliation, and prompt updates—are implemented through LLM prompting. While practical, this makes the method closer to a structured LLM pipeline than a well-specified optimization algorithm, and the RL analogy remains largely conceptual rather than algorithmically grounded.

3. **Some important design choices are under-specified.**
   In particular, the definition of **sub-trajectories** is not clearly described. Since the credit assignment step operates on these segments, the segmentation strategy may significantly influence which behaviors receive credit or blame. The paper does not clarify how trajectories are partitioned in practice (e.g., agent turns, communication rounds, or task stages), nor whether the segmentation rule is shared across tasks or dataset-specific.

4. **The aggregation mechanism is simple but potentially limiting.**
   Edits are selected based on frequency across tasks. While this stabilizes optimization, it may favor frequently occurring but low-impact edits while overlooking rarer but more impactful changes. Incorporating impact-aware or uncertainty-aware aggregation could make the update rule more principled.

5. **Experimental reporting could be more rigorous.**
   The results are encouraging, but the evaluation lacks variance reporting (e.g., standard deviations or multi-run statistics), which is important given the multiple stochastic LLM components in the pipeline. In addition, some training splits are very small, making it difficult to assess robustness and potential variance in optimization outcomes.

---

> ### Author Rebuttal · Authors · 2026-03-31
>
> We thank the reviewer for the detailed and constructive feedback. We address the main concerns on attribution validity, aggregation, and experimental rigor with additional analyses.
>
> **(1) Attribution validity (intervention-based evidence).**
>
> We agree that attribution quality should be directly validated. To address the lack of counterfactual/intervention-based validation, we perform intervention-style analyses by removing high-support edits from the optimized prompt. On the Research domain, removing top-1 / top-3 addition suggestions reduces TS from 83.0 → 82.0 / 81.33 and CS from 88.5 → 87.13 / 86.97. Similarly, removing top revision/removal suggestions reduces TS to 81.67 / 80.67 and CS to 87.71 / 87.30. These controlled interventions show that removing attributed edits consistently degrades performance, providing direct evidence that TRUCE identifies behaviorally important (causally relevant) prompt components.
>
> **(2) Aggregation mechanism.**
>
> We agree that frequency-based aggregation is simple. To validate it, we compare frequency-based ranking with optimizer confidence-based ranking and observe strong agreement (Spearman ρ≈0.78 overall, ρ≈0.80 for addition suggestions). This indicates that frequency acts as a reliable low-variance proxy for edit importance, while stabilizing noisy LLM signals across trajectories.
>
> **(3) Experimental rigor and variance.**
>
> We now report multi-run statistics on both a collaborative (Research) and a competitive (Bargaining) domain. TRUCE consistently improves over baselines with low variance (e.g., Research TS std ≈0.50 vs. 0.25–0.63; Bargaining TS std ≈0.80 vs. 0.53–0.93). Improvements remain stable across runs. Although the number of training instances is small, each instance provides rich supervision through long interaction trajectories with multiple sub-trajectories. In addition, aggregation retains only edits that are consistently supported across multiple trajectories, making it unlikely for instance-specific noise to dominate. Together, these factors reduce sensitivity to small training splits.
>
> **(4) Sub-trajectory definition.**
>
> Sub-trajectories are defined as protocol-level interaction units. For MultiAgentBench, we segment by communication rounds (coherent interaction blocks), while for the programming benchmark we use reasoning–action/task-stage blocks. These definitions follow the underlying MAS protocols and are applied consistently within each benchmark. The use of two benchmarks with different interaction structures further supports that TRUCE is not tied to a specific segmentation scheme.
>
> **(5) Policy unit construction.**
>
> Policy units are not fixed manual annotations. Instead, they are automatically constructed and updated during optimization as semantic, rule-like behavioral components. New units can be introduced, revised, or removed based on trajectory-level evidence and aggregated edit suggestions, while maintaining behavioral independence and actionability.
>
> **(6) “LLM-driven heuristics”.**
>
> While LLMs instantiate the attribution and proposal modules, TRUCE is not merely a heuristic wrapper. Instead, it defines a formal local optimization algorithm over a discrete space of policy units. The framework restricts the prompt-editing search space to a specific neighborhood of evidence-backed updates (trajectory decomposition) and applies an objective acceptance filter (cross-task aggregation). Furthermore, the modularity of this design means the LLM components could be swapped for learned or rule-based attribution mechanisms in future work without altering the underlying algorithmic structure.
>
> **(7) Optimization cost.**
>
> We appreciate the request for a clearer cost breakdown. TRUCE operates as an offline optimization procedure, where the dominant cost arises from processing long multi-agent trajectories. On the Research domain, a full optimization process for TRUCE requires 1,526 LLM calls and \~7.5M total tokens. In comparison, DSPy requires significantly more resources (3,470 LLM calls and \~14.3M tokens) due to its search-heavy process. TextGrad is more lightweight (\~628 calls and \~2.2M tokens), but this is because it applies simpler global updates without performing explicit multi-agent trajectory decomposition or credit attribution. Overall, TRUCE strikes a highly effective balance: it provides fine-grained, attribution-guided optimization at roughly half the computational cost of DSPy. This cost is incurred offline and amortized over deployment.
>
>
> We hope these additional results and clarifications address the reviewer’s concerns.

---

> > ### Author Rebuttal · Reviewer_REfD · 2026-04-04
> >
> > Thank you for the careful rebuttal. It addresses several concerns, but the reviewer does not think the central issues are fully resolved. Most importantly, the intervention analysis shows that the final selected edits are useful, but it still does not validate whether the attributed sub-trajectories actually contain the behavioral evidence that justifies those edits. The clarification on sub-trajectories and policy units also remains too high-level to make the method reproducible or to assess sensitivity to key design choices. More broadly, the reviewer remains unconvinced by the claim that TRUCE is a well-specified optimization algorithm, since its core steps still appear to rely primarily on prompted LLM judgments. Finally, the aggregation issue is only partially addressed, as the reported correlation between frequency-based and confidence-based rankings does not establish that frequency is an appropriate proxy for edit importance.

---

> > > ### Author Response · Authors · 2026-04-07
> > >
> > > We thank the reviewer for the thoughtful follow-up and for clarifying the remaining concerns. We address each point concisely and will incorporate these clarifications into the final version to ensure full reproducibility.
> > >
> > > ### (1) Attribution validity (sub-trajectory level intervention)
> > >
> > > We agree that attribution should be validated at the sub-trajectory level. We therefore performed an ablation where we masked varying sub-trajectories from the MAS system during the credit assignment phase. We compared removing 20% of (i) top-attributed, (ii) randomly selected, and (iii) low-/mid-attributed sub-trajectories.
> > >
> > > On the Research domain (original: TS = 83.00, CS = 88.50):
> > >
> > > - Removing **top-attributed** segments causes the **largest degradation**
> > >   (TS: 83.00 → 81.45; CS: 88.50 → 80.83).
> > > - **Random removal** results in a **moderate drop**
> > >   (TS: 83.00 → 81.33; CS: 88.50 → 84.99).
> > > - Removing **low/mid-attributed** segments leads to **small or non-systematic changes**
> > >   (e.g., TS: 83.00 → 83.33; CS: 88.50 → 83.85).
> > >
> > > This provides **direct intervention-based evidence** that TRUCE's top-attributed sub-trajectories genuinely capture the **critical behavioral evidence** required for successful outcomes, rather than just acting as a springboard for useful edits.
> > >
> > > ### (2) Sub-trajectory specification
> > >
> > > To make the method reproducible, we clarify that TRUCE partitions trajectories systematically based on the intrinsic execution structure of the specific MAS codebase (2nd paragraph of Section 4.2).
> > >
> > > - In **MultiAgentBench**, agent interactions are organized by a communication graph; accordingly, we take each complete **communication round** (i.e., one full exchange among agents) as the sub-trajectory.
> > > - In **CODESIM**, the framework follows a predefined sequential workflow; therefore, we take each distinct **task stage** (e.g., planning, coding, or debugging) as the sub-trajectory.
> > >
> > > This straightforward segmentation is applied consistently across all instances within each benchmark, avoiding complex heuristics and ensuring the process is easily reproducible. **Importantly, these boundaries are not manually engineered or post-hoc selected to maximize performance. Rather, they rely on native execution delimiters already present in the MAS environment, ensuring that TRUCE can generalize to new frameworks without requiring human re-segmentation.**
> > >
> > > ### (3) Policy unit specification
> > >
> > > Policy units are not reconstructed from scratch or used to replace the original prompt. Instead, the original prompt (e.g., base agent roles and instructions) is **preserved as a fixed foundation**. TRUCE performs incremental updates by introducing and modifying additional rule-like behavioral components.
> > >
> > > Policy units are dynamically constructed during optimization as interpretable rules grounded in trajectory-level evidence:
> > >
> > > - New units can be added to capture missing behaviors.
> > > - Existing units (including previously added ones) can be revised or removed based on aggregated feedback.
> > >
> > > All edits are grounded in attributed sub-trajectories and applied through **explicit add, revise, or remove operations**.
> > >
> > > ### (4) On “LLM-driven heuristics”
> > >
> > > We clarify that TRUCE is **not claimed as a fully formal optimization algorithm in the paper**. Instead, it defines a **structured and well-specified optimization procedure** over prompt space. The RL analogy is intended purely as an **intuitive perspective**.
> > >
> > > While LLMs are used to instantiate individual steps, the overall procedure is explicitly defined and fixed. TRUCE constrains the update process in three ways:
> > >
> > > - **(i) edits operate on explicit policy units;**
> > > - **(ii) edits are grounded in attributed sub-trajectories;**
> > > - **(iii) updates are selected via cross-trajectory aggregation.**
> > >
> > > The pipeline defines a constrained refinement process, where candidate updates are restricted to local, evidence-supported modifications and filtered by cross-task consistency. In this sense, the optimization is defined at the procedural level.
> > >
> > > ### (5) Aggregation mechanism
> > >
> > > We agree that frequency is not a perfect proxy for importance. Instead, aggregation acts as a **variance reduction mechanism**: it retains edits with consistent support across trajectories and filters out unstable suggestions. Empirically, this leads to stable improvements with low variance.
> > >
> > > We note that the main contribution of TRUCE lies in the **integration of trajectory-level attribution, unit-level editing, and cross-task aggregation into a coherent optimization framework**, rather than in any single aggregation rule. Within this framework, frequency-based aggregation provides a simple and effective design choice for stabilizing updates under noisy feedback in this setting. We agree that incorporating impact-aware or uncertainty-aware aggregation, as suggested, is a promising direction to further improve this component.

---

### Official Review · Reviewer_s2S9 · 2026-03-13

**Soundness:** 3
**Presentation:** 3
**Significance:** 3
**Originality:** 3
**Overall Recommendation:** 4
**Confidence:** 2

**Summary:**

This paper proposes TRUCE, a trajectory-based prompt optimization framework for LLM-based multi-agent systems. It aims to address the credit assignment problem in this setting, where final outcome feedback is too coarse to identify which parts of long, noisy interaction trajectories and which prompt rules are responsible for success or failure.

**Compliance With Llm Reviewing Policy:**

Affirmed.

**Final Justification:**

This paper is intuitive and empirically solid. My concerns are addressed with rebuttal, so I keep my positive score.

**Key Questions For Authors:**

* How sensitive is TRUCE to the choice of optimizer LLM? For example, what happens if a weaker optimizer model is used? This would help me better understand how dependent the method is on the strength of the optimizer model.

* How robust is the method to different prompt unit granularities?

* Could the authors provide a clearer analysis of the offline optimization cost, e.g., wall-clock time, number of optimizer calls, and total token usage? This would be useful to understand the practical cost of applying TRUCE.

**Limitations:**

Yes

**Strengths And Weaknesses:**

**Strengths**

* The work is clear and easy to understand. The overall pipeline of trajectory decomposition, credit attribution, rule-level edit suggestion, and cross-task aggregation is well motivated.

* The experimental results are overall strong. TRUCE consistently outperforms Original, Reflexion, DsPy, and TextGrad across multiple domains and across both GPT-4o and Qwen model families.

* The problem setting is practical, since in many real multi-agent systems the model and roles are fixed, making prompt refinement the main controllable interface.

**Weaknesses**

* The whole pipeline relies heavily on LLM-based modules, including trajectory analysis, credit attribution, suggestion generation, aggregation, and edit application. The paper does not sufficiently analyze how sensitive the method is to the quality or choice of the optimizer LLM.

* The robustness to prompt unit definition is not fully studied. Since the granularity of prompt units may directly affect both attribution quality and edit stability, more analysis here would strengthen the paper.

---

> ### Author Rebuttal · Authors · 2026-03-31
>
> We thank the reviewer for the positive assessment and helpful suggestions. We provide additional analysis on optimizer sensitivity, unit robustness, and optimization cost.
>
> **(1) Sensitivity to optimizer LLM.**
>
> We agree that sensitivity to the optimizer model is important to understand. To isolate this factor, we evaluate a self-evolving setting, where the same model (GPT-4o-mini) is used for both optimization and execution. In this setting, TRUCE still improves over the non-optimized baseline (TS: 80.00 → 80.66; CS: 84.69 → 92.40). Using a stronger optimizer (GPT-4o) further improves TS (80.66 → 83.00), while CS remains strong (92.40 → 88.50). This difference suggests that stronger optimizers tend to prioritize task completion, while weaker optimizers may better preserve coordination behaviors, leading to different trade-offs between TS and CS. Overall, TRUCE provides consistent gains even without a stronger optimizer, indicating that improvements are not solely due to optimizer strength but arise from the trajectory-aware, unit-level optimization process.
>
>
> **(2) Robustness to prompt unit granularity.**
>
> We agree that unit granularity may influence attribution. In our implementation, policy units are parsed at the level of distinct behavioral rules (e.g., individual bullet points or isolated constraints). Conceptually, if a unit is too granular or too broad, the resulting edit suggestions will be noisy. However, TRUCE's cross-task aggregation mechanism naturally mitigates this: it acts as a filter, ensuring that only edits with consistent support across multiple trajectories are applied. Empirically, our 'w/o unit refine' ablation (in Fig 5, 6), which approximates the extreme case of a monolithic, un-chunked prompt, leads to clear performance degradation, demonstrating that enforcing structured unit-level granularity is important for stable optimization.
>
> **(3) Optimization cost analysis.**
>
> We appreciate the request for a clearer cost breakdown. TRUCE operates as an offline optimization procedure, where the dominant cost arises from processing long multi-agent trajectories. On the Research domain, a full optimization process for TRUCE requires 1,526 LLM calls and \~7.5M total tokens. In comparison, DSPy requires significantly more resources (3,470 LLM calls and \~14.3M tokens) due to its search-heavy process. TextGrad is more lightweight (\~628 calls and \~2.2M tokens), but this is because it is fundamentally designed for single-agent optimization; it treats the multi-agent system as a monolithic black box, applying global gradient-style updates without performing explicit multi-agent trajectory decomposition or credit attribution. Overall, TRUCE strikes a highly effective balance: it provides the necessary, fine-grained multi-agent credit assignment at roughly half the computational cost of DSPy.
>
>
> **(4) On LLM-based pipeline.**
>
> We acknowledge that TRUCE relies on LLM modules for attribution and suggestion generation. However, the framework defines a structured local optimization process over policy units, where LLMs act as proposal and scoring components. The design is modular and can incorporate alternative attribution mechanisms, which we view as an important direction for future work.
>
> We hope these additional analyses clarify the robustness and practicality of TRUCE.

---

> > ### Author Rebuttal · Reviewer_s2S9 · 2026-04-03
> >
> > Thanks for the rebuttal. My concerns are addressed.

---

### Decision · Program_Chairs · 2026-04-30

**Decision:**

Accept (regular)

**Comment:**

The paper proposes TRUCE a framework for prompt optimization in multi-agent systems. The main problem that it solves is credit-assignment, which is used to do prompt refinement.

Reviewers generally found the problem important, proposed approach as intuitive, and experimental evidence to be good.

Main weakness raised were:

1. Heavy reliance on LLM without adequate ablations to know sensitive the performance is to the choice of LLM. Authors have evaluated with a weaker optimizer (GPT-4o-mini) and shown that it can still improve the performance.

2. Comparison is made against single-agent prompt opt methods and not recent multi-agent ones. Authors have added the two recent work MAPRO, EACL’26; HiveMind, AAAI’26 and shown that while these methods are strong, the proposed approach outperforms them. This is a strong claim, if fairly evaluated.

3. Sub-trajectory/prompt unit/instances are not adequately defined. Authors have added details of these.

4. Quality of credit assignment is not directly evaluated. Authors have added experiments showing that removing the top-attributed segments leads to the biggest degradation.

5. Reporting variance/confidence interval will be nice. Authors have added these in the rebuttal.

Overall, rebuttal has significantly strengthened the paper. Reviewer s2S9  and vmHa found their concerns fully addressed, and while reviewer JGaW did not respond to the rebuttal, I think it adequately addresses their main concerns. Many concerns of reviewer REfD are also addressed specially with additional experiments. Therefore, I lean towards accepting this paper.